# Circular RNA- and microRNA-Mediated Post-Transcriptional Regulation of Preadipocyte Differentiation in Adipogenesis: From Expression Profiling to Signaling Pathway

**DOI:** 10.3390/ijms24054549

**Published:** 2023-02-25

**Authors:** Chiu-Jung Huang, Kong Bung Choo

**Affiliations:** 1Department of Animal Science & Graduate Institute of Biotechnology, School of Agriculture, Chinese Culture University, 11114 Taipei, Taiwan; 2Department of Preclinical Sciences, M Kandiah Faculty of Medicine and Health Sciences, Universiti Tunku Abdul Rahman, 43000 Selangor, Malaysia

**Keywords:** circular RNA, microRNA, post-transcriptional regulation, signaling pathways, preadipocyte differentiation, adipogenesis, species conservation

## Abstract

Adipogenesis is an indispensable cellular process that involves preadipocyte differentiation into mature adipocyte. Dysregulated adipogenesis contributes to obesity, diabetes, vascular conditions and cancer-associated cachexia. This review aims to elucidate the mechanistic details on how circular RNA (circRNA) and microRNA (miRNA) modulate post-transcriptional expression of targeted mRNA and the impacted downstream signaling and biochemical pathways in adipogenesis. Twelve adipocyte circRNA profiling and comparative datasets from seven species are analyzed using bioinformatics tools and interrogations of public circRNA databases. Twenty-three circRNAs are identified in the literature that are common to two or more of the adipose tissue datasets in different species; these are novel circRNAs that have not been reported in the literature in relation to adipogenesis. Four complete circRNA–miRNA-mediated modulatory pathways are constructed via integration of experimentally validated circRNA–miRNA–mRNA interactions and the downstream signaling and biochemical pathways involved in preadipocyte differentiation via the PPARγ/C/EBPα gateway. Despite the diverse mode of modulation, bioinformatics analysis shows that the circRNA–miRNA–mRNA interacting seed sequences are conserved across species, supporting mandatory regulatory functions in adipogenesis. Understanding the diverse modes of post-transcriptional regulation of adipogenesis may contribute to the development of novel diagnostic and therapeutic strategies for adipogenesis-associated diseases and in improving meat quality in the livestock industries.

## 1. Introduction

### 1.1. Adipogenesis Is a Crucial Cellular Process

Adipose tissue mass may expand via increasing the size of the constituent adipocyte cells. On the other hand, adipogenesis is an adipocyte biogenesis process in which new adipocytes are generated from multipotent progenitor stem cells. Adipogenesis begins with the progenitor cells being committed to become preadipocytes, which undergo growth arrest, followed by preadipocyte differentiation into mature adipocytes [1,2].

Adipose tissues are morphologically divided into the white (WAT) and brown adipose tissue (BAT), and the beige adipose tissue, with each type playing a different physiological role. Morphologically, a mature WAT adipocyte carries a large lipid droplet but few mitochondria within the cell. Hence, WAT serves mainly as an energy reservoir; excessive lipids are stored as triglyceride, which is converted, on energy demand, to free fatty acids for circulation [3,4,5]. Dysfunctional WAT is associated with obesity, insulin-resistance in diabetes, cardiovascular disorders and cancers, among other human conditions [6,7,8,9]. On the other hand, small lipid droplets and high numbers of mitochondria are found in BAT adipocytes. Functionally, BAT serves a thermogenic function in producing heat from the lipid droplets, regulating body temperature, in addition to other secretory functions [10,11,12]. While excessive WAT in obesity is unhealthy, BAT is favorable in health in its effects on reducing the accumulation of adipose tissues and in lowering insulin resistance in diabetes patients (reviewed in [13]). Beige adipocytes may be converted from WAT adipocytes, a process dubbed WAT browning. Beige adipocytes possess smaller oil droplets and enriched mitochondria contents; hence, beige adipose tissues share many features and physiological functions with BAT [14]. WAT browning is typically induced by exposure to cold and on demand for heat [15].

In term of regulation, a key gateway to adipogenic gene expression that drives preadipocyte differentiation is the peroxisome proliferator-activated receptor (PPAR) family nuclear proteins, particularly the PPARγ isoform [16,17,18,19,20]. In several human clinical conditions, including inflammation, insulin sensitivity, obesity and cancer, PPARγ has been shown to play important causative roles (reviewed in [18]). In thermogenesis, there is also crosstalk between PPARs and thyroid hormone receptors in the adipogenesis process via competing for binding to the heterodimeric partner, retinoid X receptor or other targets [21,22,23]. Hence, adipogenesis, acting through PPARs or otherwise, is a crucial cellular process for human wellbeing and survival.

In animal husbandry, studies on adipogenesis are focused on improving the quality and nutritional values of meat of livestock, which are a major source of proteins and lipids for humans. A major factor that affects meat quality is intramuscular fat content, which controls not only meat texture and taste, but also supplements of essential fatty acids [24]. Hence, extrapolations from animals to the human, and vice versa, may rapidly help expand understanding in the regulation of adipogenesis and the adipogenesis-associated proteins across species. The assumption is that if a transcript or protein sequence is highly conserved through evolution, critical biological functions are implied. Interspecies sequence conservation has, indeed, been useful in identifying causes of congenital diseases in humans [25,26]. Based on this supposition, we have previously reviewed possible extrapolation of species-conserved microRNAs (miRNAs) and the miRNA-targeted mRNAs of chicken in adipogenic gene expression in the adipogenesis process [16].

### 1.2. CircRNA- and miRNA-Mediated Post-Transcriptional Regulation of Gene Expression: An Overview

Studies have shown that adipogenesis is regulated by a wide array of genes [17]. On reaching the cytoplasm, mature mRNAs may further be modulated post-transcriptionally by regulatory RNA species, including circular RNA (circRNA) and microRNA (miRNA), in deciding “go or no go” in translation into a functional protein (Figure 1). Translation of an mRNA may be blocked by a targeting miRNA, or the mRNA may suffer degradation induced by the targeting miRNA (Figure 1B). The miRNA is, in turn, under the whip of another single-stranded but larger-sized circular RNA (circRNA) via base-pairing resulting in the “sponging” off, i.e., in depleting, of the miRNA pool to free the targeted mRNA for translation (Figure 1C). Subsequently, the translated protein goes into signal transduction or other biochemical pathways to regulate adipogenic gene expression (Figure 1A–C) [16,27]. Clinical studies have indicated that circRNAs and miRNAs may be encapsulated in exosomes, or simply released as free forms into the blood stream of patients to be transported to destination cells, which may or may not express the circRNA or miRNA, to exert gene regulatory functions in the destination cells (Figure 1D) [28].

It is noted here in passing that the long noncoding RNAs (lncRNAs) constitute another unique group of noncoding RNAs that interact with other biomolecules, including circular RNAs and microRNAs, to affect biological functions at multiple regulatory levels [29,30,31]. However, lncRNAs are not the focus of this work, and are discussed only in relation to their interactions with the circRNA and miRNA networks being analyzed.

The biogenesis of and functions of miRNA have been extensively reviewed [32,33,34,35]. Only a synopsis of key features relevant to this review is given here. Each miRNA is encoded by a single gene, which may occur in clusters; an extensively studied example is the chromosome 19 miRNA cluster, or C19MC, which includes 46 individual miRNA genes [36,37]. Evolutionary transposition has also generated miRNA families, each with multiple family members with identical or highly homologous sequences; a notable example is the let-7 miRNA family [38,39]. It is noteworthy that many miRNA genes are located within the intron sequences of many protein-coding genes, a genomic framework also shaped by evolution [40,41]. In the process of miRNA biogenesis, a hairpin precursor is first formed, which is further processed to generate either or both the 3- or 5-prime (3p or 5p) mature miRNA species with different sequences targeting different mRNAs in action (Figure 1B) [42,43]. Hence, the suffix -3p or -5p is important in designing a specific miRNA to avoid confusion. The soul of the 17–21-nucleotide miRNA is the “seed sequence” of 5–8 nucleotides located at the 5′-end of the short RNA sequence. Members of the same miRNA family share identical seed sequences. In targeting an mRNA, the miRNA seed sequence interacts with a complementary sequence in the 3′-untranslated region (3′-UTR) of the mRNA; the sequences on both sides of the seed regions often show low sequence homology without affecting the mRNA-targeting action of the miRNA.

The biogenesis of circRNAs is more elaborate [44]. Unlike miRNA, which each have a specific-gene status, a circRNA is the backsplicing offspring of a pre-mRNA of a specific host gene by stitching together one or more selected exons and/or intron sequences of the pre-mRNA into a circular RNA molecule of assorted sizes (Figure 1C). Backsplicing requires canonical pre-mRNA spliceosomal machinery, recognizing the same consensus intron–exon GU-AG junctions. The process is also modulated by the pairing of intronic repetitive sequences, such as *Alu*, and contributions by cis complementary sequences and trans-acting protein factors [45,46]. Multiple circRNA species, or isoforms, may be generated from a specific host transcript. Different isoforms carry different sequences and, therefore, sponging off of different miRNA targets. In the literature, the circRNA nomenclature has not been standardized, and is, therefore, rather confusing. A circRNA designation may be provided as: (i) a circBase (http://www.circbase.org, accessed on 20 December 2022) ID, which is specific for each circRNA, but does not provide any hint of the isoform on first sight; (ii) the host gene name but often without indication of the isoform being studied; or (iii) the chromosomal location of the circRNA sequence, which is not reader friendly. In this review, efforts are put into identifying both the circBase ID and the host gene name of the reported circRNAs being discussed.

We have previously reviewed the miRNA–mRNA signaling axis that impacts the PPARγ gateway in the adipogenesis process [16]. In this review, we are extending the literature appraisal to include circRNAs in the post-transcriptional regulatory network, focusing on circRNA–miRNA and miRNA–mRNA crosstalk in regulating preadipocyte differentiation. The approach used is integration and analysis of information and datasets harvested from literature scrutiny, interrogation of circRNA and miRNA databases and analysis of the acquired information using bioinformatics and algorithmic tools.

## 2. Whole-Transcriptome Profiling of circRNAs in Preadipocyte Differentiation in Adipogenesis: Connecting the Human and Animal Species

### 2.1. Profiling of Adipogenesis-Associated circRNAs in Different Species: Methodology and Dataset Availability

To uncover novel circRNAs associated with adipogenesis, whole-transcriptome profiling has been applied in humans and assorted experimental and domesticated farm animals. Reports of such studies are obtained through searches in the PubMed (https://pubmed.ncbi.nlm.nih.gov, accessed on 20 December 2022) database. Many studies are focused on the critical step of differentiation of preadipocytes into mature adipocytes by examining differential expression (DE) data before and after differentiation induced in vitro (Table 1A). Besides preadipocyte differentiation, comparative studies on subjects relevant to adipogenesis are also presented, e.g., circRNA profiling in adipocytes between obese and lean individuals, and between developmental stages in calf and adult cattle (Table 1B) [47,48]. We also include here a paper that does not have DE data but provides useful datasets on visceral (VAT) and subcutaneous adipose tissues (SAT) in humans and mice (Table 1B, rows B2 and B3) [49]. In total, 11 studies with 12 datasets that encompass the human and six animal and bird species are included in this review (Table 1).

In early circRNA profiling studies, a human circRNA microarray platform was used, which generated non-discriminating and not-user-friendly datasets [47,50]. Subsequently, most circRNA profiling works used the high-throughput whole-transcriptome RNA sequencing (RNA-seq) platform. The quality of the profiling datasets obtained are affected by the RNA preparations used in the RNA-seq work. In a study in which total RNA preparations were used in a microarray platform, a high number (4080) of differentially expressed (DE) circRNAs are reported, while the total number of circRNAs was not revealed (Table 1A, row A1) [50]. In most studies, ribosomal RNA-free RNA preparations were used. In a few studies, circRNA was further enriched by RNase R treatment to remove other linear RNA species [47,51,52]. The RNA preparations were then used in constructing unidirectional strand-specific RNA libraries for RNA-seq, followed by data analysis using appropriate algorithms and bioinformatics tools. In most cases, the circRNA expression datasets, after various degrees of annotation and organization, were submitted either as Appendix A for online access on publication of the papers, or were deposited in some public databases, including miRbase (https://www.mirbase.org, accessed on 20 December 2022) and miRDB (https://mirdb.org, accessed on 20 December 2022). In this review, the availability of useful circRNA profiling datasets is indicated (Table 1).

For whole-transcriptome profiling, the availability of useful and user-friendly datasets is important for comparative analysis. Many authors provide circRNA names either in the host name nomenclature, e.g., circSAMD4A, or as circBase identification (ID) numbers, e.g., hsa_circ_0004846. However, authors, except Arcinas et al. (2019) [49], often neglect to be more specific when different circRNA isoforms are detected [49]. Other useful circRNA identification information includes the GenBank (https://www.ncbi.nlm.nih.gov/genbank/, accessed on 20 December 2022) accession number of the host gene transcript from which the circRNA is derived, the exons retained in the circRNA and the chromosomal positions of the retained exons [49]. In many circRNA profiling papers, only circBase ID numbers are provided, which make it difficult to prompt the identity of the circRNA concerned. Hence, we have put in efforts in this review on better identifying the specific host genes and translating the circBase ID into the host gene nomenclature whenever is possible. In some works, the supplementary circRNA datasets provided are in chromosomal positions of the predicted retained exons without the putative circRNA IDs or host gene names, which makes cross referencing difficult [53,54]. In some cases, accessible and useful datasets are unavailable (Table 1).

### 2.2. Only a Small Number of circRNAs Are Involved in Preadipocyte Differentiation: Findings from Profiling Studies

Discounting the microarray-based work on human by Sun W et al. (2020) [50], the total number of circRNAs expressed in the adipocytes in different species ranges from 2172 in pig to 7203 in yak (Table 1A) [53,54]. On differentiation, the fraction and the number of differentially expressed (DE) circRNAs in the mature adipocytes relative to the preadipocytes before differentiation ranges from 1.09% (41 circRNAs) to 3.02% (117) in the stromal cells of mouse WAT and BAT, respectively (Table 1A, rows A2 and A3). A much higher, but seemingly unrealistic, DE number of 4080 was observed when total RNA was used in the microarray platform [50]. In a study in pig in which lncRNA and mRNA were also included in the analysis [54], a high fraction (13.67%, 297 circRNAs) of DE circRNA was also reported, probably due to the use of different algorithms in analyzing the sequencing datasets. Hence, these two studies are excluded from further analysis.

Taken together, the small number of DE circRNAs may indicate that only a small number of circRNAs participate in the preadipocyte differentiation process (Table 1A). Furthermore, DE circRNAs may be up- or downregulated, suggesting that circRNAs may act as positive or negative modulators in modulating downstream adipogenic gene expression.

Besides differentiation, four comparative studies, generating six datasets, on adipogenesis-related subjects have been included (Table 1B). In the work comparing human obese vs. lean subjects, 244 DE circRNAs are revealed (Table 1B, row B1) [47]. In comparing two breeds of pig with different fat contents, 275 circRNAs, mostly downregulated, are reported (Table 1B, row B4) [55]. Further examination of the DE circRNAs presented in these two works may contribute to producing slimmer persons in the clinic and leaner farm animals in the meat industry. In development-related adipogenesis, 67 (1.38%) DE circRNAs are reported between young and old rats and 307 (7.08%) in calf and adult cattle (Table 1B, rows B5 and B6) [48,56].

### 2.3. Preadipocyte Differentiation-Associated circRNAs in the Adipose Tissue: Extrapolating Animal Data to the Human

A comprehensive and informative profiling work on preadipocyte differentiation in mouse WAT-1 is presented by Zhang P.P. et al. (2021) [51] (Table 1A, row A2). In the study, 28 circRNAs are identified as upregulated and 13 as downregulated on differentiation, including four circRNAs, each with two isoforms (Table 2, row 1). Based on the log_2_fold changes of these 41 DE circRNAs provided in this work, comparative assessments are made with other WAT and BAT circRNA datasets in mouse, human and yak, reported by other authors (Table 2, rows 2–5; Appendix A). When compared with the mouse WAT-2 dataset of Arcinas et al. (2019) [49], only 25 (61.0%) of the 41 circRNAs in the Zhang P.P. et al. (2021) [51] dataset are found in both mouse datasets, including 20 up- and 5 downregulated in expression on differentiation (Table 2, rows 1 and 2; Appendix A). The 25 circRNAs may be considered as validated circRNAs that are involved in preadipocyte differentiation in the mouse WAT. Interestingly, PubMed interrogations indicate that none of these circRNAs have been reported before in relation to adipogenesis, indicating that they are novel circRNAs awaiting further investigation into their regulatory role in adipogenesis. These circRNAs are not further discussed here.

In the human and yak WAT, 23 (56.1%) and 19 (46.3%) circRNAs are in common in the mouse WAT-1 dataset, respectively (Table 2, rows 3 and 4; Appendix A). The common circRNAs that appear in the WAT datasets of mouse, human and yak may be interpreted as that these circRNAs that are conserved in mandatory functions in modulating preadipocyte differentiation in different species. Thirty-two (78.0%) circRNAs, including three isoforms, are commonly expressed in WAT and BAT differentiation in the different species analyzed (Table 2, row 5; Appendix A), indicating common circRNA regulatory pathways in WAT and BAT differentiation. It is also noted that 9 of the 41 mouse WAT-1 circRNAs are WAT-specific and are not detected in mouse BAT (Appendix A), suggesting that some regulatory pathways are exclusive to WAT differentiation. The nine circRNAs are chr17:34877211-34956589, Cacna1d and Fancl in the upregulated group and Rad18, Megf8, Trpc6, Zfp532, Dcbld2, Zfx in the downregulated group. The comparative analysis presented here is based on a limited number of datasets and species analyzed. The predictions made should be taken with caution pending further confirmation of their involvement in adipogenesis differentiation.

## 3. Cross-Species Conservation of the circRNA–miRNA and miRNA–mRNA Interacting Sequences in Adipogenesis Differentiation

In two RNAseq profiling studies (Table 1B, rows B1 and B5), the circRNA–miRNA–mRNA trio association, validated in luciferase assays, are reported: circSAMD4A-miR-138-5p-EZH2 mRNA in the human and the circFUT10-let-7c-5p-PGC1β/PPARGC1B in the cattle; circRNA-mediated preadipocyte differentiation was also demonstrated in both cases (Table 3) [47,48]. Further PubMed searches were conducted using the stringent criteria of validation of circRNA–miRNA and miRNA–mRNA interactions by luciferase and mutational or pulldown analysis. Two other circRNAs are identified: the bovine bta_circ_Pparγ and bta_circ_Flt1 (Table 3). Taken together, four circRNAs with established circRNA–miRNA–mRNA connections are found in one or more of the general WAT and BAT circRNA datasets of human, mouse and yak presented in Table 1 and Table 2 above (Table 3). However, they are not among the list of 41 differentially expressed circRNAs in the mouse dataset, most likely because of the imposed criteria constraints in the identification of these circRNAs, and possibly because of species differences. The basic molecular features of the four selected circRNAs, including the exons of the host transcript, retained, size, transcript ID of the host transcript and chromosomal positions are shown in Appendix A. It is noted that circPPARγ- and circFLT1-modulated miR-92a-3p and miR-93-5p affect more than one mRNA species and, therefore, different cellular processes, and that two long noncoding RNAs also participate in the circFLT1-miR-93-5p regulation (Table 3; see below).

### 3.1. miRNA Conservation

Sequence conservation in modulatory RNAs across species is a good indicator of the importance of the regulated cellular functions. If found highly conserved, similar action and function may be predicted across species [16]. Since miRNA plays a central role in connecting circRNA to mRNA, the miRNAs in the validated interactions described above are first examined (Table 3). It is first noted that each of the four miRNAs belongs to a specific miRNA family, and that the genes of miR-92a-3p and miR-138-5p are found in two different chromosomal clusters (Table 3; Appendix A). Since members of the same family share identical seed sequences, family members may target the same transcript and share regulatory functions. On the other hand, appearance in chromosomal clusters is an indication of active sequence evolutionary histories [63,64].

The miRNA sequences of the human, mouse, rat, pig, bovine and chicken obtained through miRBase and TargetScan (https://www.targetscan.org, accessed on 20 December 2022) interrogations are aligned (Figure 2A and Table 4). Except for the unavailability of the pig and chicken miR-93-5p sequences, the 6–7-nucleotide seed sequences of all four adipogenesis-associated miRNAs are identical in the six species analyzed, including the avian chicken. Moreover, the non-seed sequences of the miRNAs are also highly conserved (Figure 2A). The observed miRNA sequence conservation supports the proposition that these miRNAs play crucial functional roles in adipogenesis across species.

### 3.2. Conservation in circRNA–miRNA Interactions

CircRNA sponging of miRNA kickstarts the regulatory role of circRNA in modulating mRNA translation and the subsequent cellular functions. For circRNA–miRNA alignments, circRNA sequences in human, mouse and bovine are obtained from the circBase and circBank (http://www.circbank.cn, accessed on 20 December 2022) databases. The analysis shows that the miRNA seed sequences align perfectly with the targeted sequences of the circRNAs (Table 4), allowing for the rare wobble G-U base-paring in the RNA species (Figure 2B(i,ii), underlined nucleotides). However, a single mismatch in the circFLT1-miR-93-5p pair and two mismatches in circFUT10-let-7c-5p in the seed sequences in the mouse are noted (Figure 2B(ii,iii), in black letters). Furthermore, the circRNA sequences outside the seed regions also show a high degree of sequence homology in the three species, further supporting species conservation of the circRNA and miRNA interactions.

### 3.3. Conservation in miRNA–mRNA Interactions

Literature scrutiny has revealed that miR-92a-3p targets the C/EBPα (CCAAT Enhancer-Binding Protein alpha) and p130/Rb2 (Retinoblastoma 2/p130) transcripts and miR-93-5p targets SIRT7 (Sirtuin-7) and TBX3 (T-Box Transcription factor 3) (Table 3). On alignment of the miRNA and mRNA sequences of interaction, all seven to eight nucleotides of the miRNA seed sequences are found to align perfectly with the complementary targeted sites in the 3′-UTR (3′-untranslated region) of the mRNAs in most cases, with only rare single-nucleotide disparity in some cases, particularly in chicken (Figure 2C; Table 4, last column). Furthermore, two miRNA target sites are found in the 3′-UTR of PGC1β/PPARGC1B (PPARγ Coactivator 1-β) mRNAs of all species analyzed, both of which are also fully conserved in the let-7c-5p seed sequence (Figure 2C(iii)). The interaction of miR-138-5p-EZH2 (Enhancer of Zeste Homolog 2) is also perfectly aligned (Figure 2C(iv)). Taken together, the high degree of miRNA seed sequence conservation in the miRNA–mRNA interactions between species predicts conservation of the modulation mechanism of adipogenesis across species.

## 4. Selected circRNA- and miRNA-Mediated Post-Transcriptional Regulation of Signaling and Biochemical Pathways in Preadipocyte Differentiation

In this section, the four selected circRNAs and the associated miRNAs and mRNAs that modulate preadipocyte differentiation are individually discussed. The salient molecular features of the four circRNAs are shown in Appendix A. Emphasis in our discussion is on the molecular events controlled by the proteins, the translation of which is regulated by the circRNAs and miRNAs. We have shown above that the circRNAs, miRNAs and mRNAs concerned are highly conserved in the interacting sequences (Table 4; Figure 3). Hence, the events are discussed in general without species specification. However, the human or animal species of the RNAs investigated in the cited studies is specified.

### 4.1. CircPPARγ Sponges miR-92a-3p to Regulate C/EBPα and p130/Rb2

On induced differentiation of bovine adipocytes, bta_circ_Pparγ (bta_circ_0010660) inhibits adipocyte apoptosis and proliferation while promoting adipocyte differentiation via the sponging of miR-92a-3p [57]. However, the authors did not identify the mRNA targeted by miR-92a-3p.

In an early study, the whole of the miR-17/92 cluster, amongst which is miR-92a (Appendix A), was used in preadipocyte differentiation studies in mouse 3T3L1 preadipocyte cells [58]. Upregulated expression of members of the miR-17-92 cluster is shown to promote the clonal expansion stage of adipocyte differentiation via targeting p130/RB2 (Retinoblastoma 2), echoing a previous report [65], and supported by our seed sequence analysis that miR-92a-3p targets a 3′-UTR sequence of the p130 mRNA, and the seed sequence is conserved in different species (Figure 2C(i); Table 4). Before terminal differentiation, differentiating preadipocytes are arrested in growth when re-entry of the cell cycle is blocked [66]. circPPARγ-induced downregulation of miR-92a-3p results in increased p130 levels to enhance p130/E2F dimerization [67] and association with the transcription factor DP-1 [66,68]. The consequence is the exit of the cell cycle, growth arrest and terminal differentiation to form mature adipocytes. In uncommitted human bone marrow adipose tissue-derived stromal cells, absence of p130 has, indeed, been shown to hamper terminal adipocyte differentiation [69]. Hence, targeting p130 is the first route by which circPPARγ exerts its influence on adipogenesis via miR-92a-3p by influencing cell cycle, growth arrest leading to terminal differentiation (Figure 3A, left panel, route I).

It has also been reported that C/EBPα, when induced in mouse 3T3-L1 preadipocytes in the early stage of differentiation, prompts p130/E2F association via p21 upregulation [59]. In this way, C/EBPα may also regulate E2F availability in the activation of the cell cycle in the clonal expansion stage, leading to terminal differentiation and formation of mature adipocytes (Figure 3A, left panel, route II).

In a different study, chronic myeloid leukemia (CML)-derived exosomes that harbor human miR-92a-3p are shown to promote adipogenesis of adipose-derived mesenchymal stem cells [60]. Our analysis supports that the seed sequence of miR-92a-3p perfectly complements a specific sequence of the 3′-UTR of the C/EBPα transcript, and that the seed sequence is conserved (Figure 2C(i); Table 4). In the same CML-exosome study, in vitro studies show that miR-92a-3p suppresses PPARγ and C/EBPα expression and, consequently, the expression of the adipogenic genes, FABP4 (Fatty Acid Binding Protein 4) and AdipoQ (Adipocyte, C1q And Collagen Domain-Containing Protein) (Figure 3A, right panel, route III) [60]. The results are consistent with a previous report that high C/EBPα levels trigger preadipocyte differentiation and WAT development [70]. The C/EBPα-C/EBPβ heterodimer often acts in concert with PPARγ to form a gateway to adipogenic expression and the maintenance of the differentiated state of adipocytes (Figure 3A, right panel, route III) [16,71,72,73]. Importantly, CML-derived exosomal miR-92a-3p is linked to induction of loss of bodyweight via WAT browning and increased energy expenses in cancer-associated cachexia [8].

### 4.2. CircFLT1 Sponges miR-93-5p to Regulate SIRT-7 and TBX3

In induced differentiation of bovine preadipocytes, miR-93, which should be miR-93-5p, is identified as the top expressing miRNA [61]. Through TargetScan analysis and luciferase assays, circFLT1 (bta_circ_002673) and the long noncoding RNAs, lncCCPG1 and lncSLC30A9, are shown to bind competitively to miR-93-5p. On the other hand, circFLT1 and lncCCPG1 also compete to deplete miRNA-93-5p from binding to lncSLC30A9 to offset the lncSLC30A9 action in inducing upregulated expression of PPARγ, C/EBPα and FABP4, leading to preadipocyte differentiation (Figure 3B, route I). The mechanism proposed by the authors is that lncSLC30A9 binds to and transports c-Fos into the nucleus to activate the PPARγ promoter, leading to differentiation (Figure 3B, route I) [61,74].

MiR-93 is a member of the miR-106b/25 cluster (Appendix A). In another study using miR-106b/25-knockout mice, miR-93-5p is shown to target SIRT-7 (Sirtuin-7) [62]. SIRT-7 is a NAD-dependent deacetylase of histones that induces transcriptional repression [75]. Since SIRT-7-knockout mice have less visceral fat, the gene is linked to adipogenesis [76]. In knockout mice, SIRT-7 is shown to deacetylate and, hence, activate another SIRT protein, SIRT-1, in the preadipocyte differentiation process (Figure 3B, route II) [77]. On the other hand, FOXO1 (Forkhead Box O1), previously inactivated upstream by being acetylated and also phosphorylated by AKT signaling, is now being re-activated by deacetylation by SIRT-1 and dephosphorylation by PP2A (protein phosphatase 2). Subsequently, the activated FOXO1 protein binds to the PPARγ promoter to block PPARγ expression in cis, or interacts with PPARγ in trans, to deplete PPARγ for utilization in adipogenesis (Figure 3B, route II) [16,78,79,80,81]. When miR-93-5p is sponged by circFlt-1, expression of SIRT-7 is upregulated, suppressing SIRT-1 expression to prevent deacetylation and re-activation of FOXO1, thus, allowing PPARγ to participate in adipogenesis.

In the same study, miR-93-5p targeting of TBX3 (T-Box Transcription Factor 3) is demonstrated, which results in suppression of self-renewal in adipocyte precursors before commitment to differentiation in the very early stage of adipogenesis (Figure 3B, route III) [62]. TBX3 has previously been shown to contribute to osteogenic differentiation of human adipose stroma cells and in maintaining pluripotency via targeting the promoter of Oct-4, one of crucial pluripotency inducers in precursor stem cells [82,83].

### 4.3. The circFUT10-let-7c-5p-PCG1β Regulatory Pathway

In RNAseq analysis, one of top differentially expressing circRNAs in adipose tissues of both young and adult cattle is circFUT10 (Table 1, row B6) [48]. In cattle, circFUT10 (circRNA ID not available) is further shown to promote adipocyte proliferation by increasing the number of adipocytes in the S and G2 phases of the cell cycle, while suppressing adipocyte differentiation in in vitro assays in bovine adipocyte cells. CircFUT10 sponges let-7c, which we show is let-7c-5p, and that let-7c-5p targets PGC1β/PPARGC1B (PPARγ coactivator 1-β) (Figure 3C). PPARγ acts in collaboration with retinoid X receptor (RXR) to bind to PPARγ response elements in the promoters’ PPARγ-modulated genes in the adipogenesis process [84,85]. PGC1β also associates with PPARγ to induce further PPARγ interactions with other transcription factors in various processes, including tumorigenesis [86,87]. In adipogenesis, however, PGC1β acts as a PPARγ repressor in adipocyte differentiation [88,89]. It is also noteworthy that PGC1β has been shown to be activated, at least in part, by PRDM16 (PR/SET Domain 16 Protein) in the earlier fate determination of brown fat adipogenesis [90]. Whether PRDM16 also activates PGC1β in preadipocyte differentiation remains to be shown.

### 4.4. The circSAMD4A-miR-138-5p-EZH2 Pathway

In a circRNA profiling work on adipose tissues in obese and lean human individuals, one of the top-expressing circRNAs in preadipocytes is circSAMD4A (hsa_circ_0004846). CircSAMD4A expression levels are correlated with bodyweight (Table 1, row B1) [47]. Knockdown of circSAMD4A downregulates expression of PPARγ and C/EBPα and inhibits preadipocyte differentiation via sponging of miR-138-5p, which releases EZH2 (Enhancer of Zeste Homolog 2) mRNA from miR-138-5p translational suppression (Figure 3D, top portion). miR-138-5p has, indeed, been shown earlier to be a negative modulator of adipogenic differentiation of human adipose-derived mesenchymal stem cells [91].

EZH2 is a histone methyltransferase that epigenetically regulates gene expression through methylation of the histone H3K27, resulting in chromatin changes [92,93,94]. Under normal circumstances, methylated H3K27 binds to the WNT promoter, thereby suppressing WNT expression [95]. Involvement of the EZH2-WNT signaling in adipogenesis has been independently reported by several laboratories [95,96,97]. The WNT-1 protein, on entering the nucleus of preadipocyte cells, downregulates expression of PPARγ and C/EBPα through β-catenin association with other transcription factors to consequently suppress preadipocyte differentiation [16,98]. In short, circSAMD4A upregulates EZH2 expression via sponging miR-138-5p to boost H3K27 histone methylation, thereby suppressing canonical WNT/β-catenin signaling and activating the PPARγ-C/EBPα gateway to advance adipogenesis (Figure 3D).

It is noteworthy that miR-138-5p and EZH2 are hot subjects in cancer research. circSAMD4A also sponges another miRNA, viz. miR-1244, which targets the transcript of ubiquitin protein ligase MDM2 in promoting proliferation and enhancement of stem cell characteristics of osteosarcoma cells [99]. Besides circSAMD4A, miR-138-5p is also targeted by lncHCP5, lncSNHG7 and lncDSCAM-AS1 in promoting tumor growth in various cancers, all of which also act through EZH2 [100,101,102]. Furthermore, besides the WNT/β-catenin signaling in adipogenesis, the miR-138-5p and EZH2 act in concert to affect other signaling pathways in the tumorigenesis process [103,104,105]. All such observations further support that there exist cross talks between adipogenesis and tumorigenesis.

## 5. Concluding Remarks

In this review, we have identified and elucidated the regulatory role of sets of circRNAs and miRNAs that modulate post-transcriptional expression of proteins directing chemical signals to the PPARγ-C/EBPα gateway and other entry points to activate adipogenic gene expression in preadipocyte differentiation. A summary of the pathway of analysis and the major findings of the four complete circRNA- and miRNA-mediated regulatory pathways leading to adipogenesis is shown in Figure 4. Part (I) involves dissection of circRNA profiling datasets, which leads to the identification of 32 novel WAT and BAT differentiation-associated circRNAs that await elucidation in their regulatory roles in adipogenesis. In part (II) of our analysis, four circRNAs and the respective interacting miRNAs and mRNAs are identified and the downstream signaling and biochemical pathways are analyzed.

A salient finding is conservation in the seed sequence of interactions in the circRNA–miRNA and miRNA–mRNA pairs, supporting that these circRNAs and miRNAs play crucial roles in post-transcriptional regulation in preadipocyte differentiation in the adipogenesis process. Sequence conservation may also justify extrapolations and projections of data between the human and animal species, pending more direct demonstration cross species, but speeding up clinical studies in the human in adipogenesis-associated diseases. No less important, elucidation of regulatory circRNAs and miRNAs in adipogenesis may also have impacts on improving meat quality in the livestock industry.

## Figures and Tables

**Figure 1 ijms-24-04549-f001:**
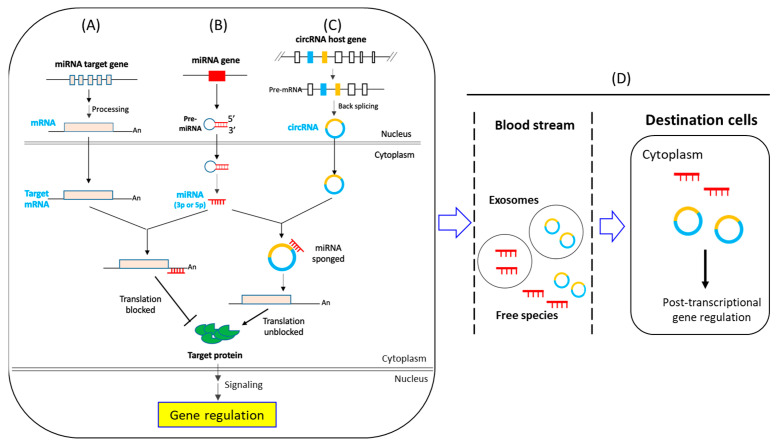
The circRNA- and miRNA-mediated post-transcriptional regulation of gene expression: an overview. (**A**–**C**) Basic mechanisms of biogenesis of mRNA (**A**), miRNA (**B**) and circRNA (**C**), and interactions between the various RNA species. Notably, the transcript of a miRNA gene first forms a precursor pre-miRNA, which matures into single-stranded 5p and/or 3p miRNA species to bind to the 3′-UTR of the targeted mRNA to induce mRNA degradation or to block translation. A mature circRNA, formed via backsplicing of the pre-mRNA of a host gene, sponges to deplete the targeted miRNA, unblocking mRNA translation by the miRNA. (**D**) CircRNA and miRNA may also be released into the blood stream encapsulated in exosomes, or as free forms, to reach a distant destination cell to exert their post-transcriptional regulatory functions.

**Figure 2 ijms-24-04549-f002:**
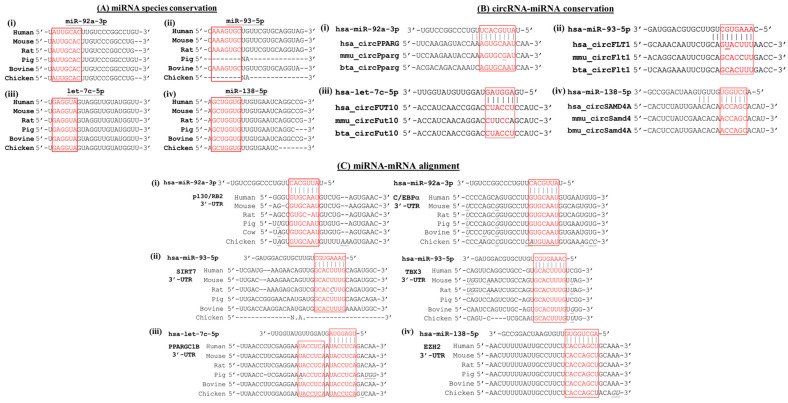
Cross-species alignment and conservation of the circRNA–miRNA and miRNA–mRNA interacting sequences in adipogenesis differentiation. (**A**) miRNA conservation among different species. (**B**) circRNA–miRNA interactions. (**C**) miRNA–mRNA interactions. circRNA sequences are derived from circBase (http://www.circbase.org) and circBank (http://www.circbank.cn); miRNA sequences are derived from miRbase; 3′-UTR of mRNA is derived from TargetScan (https://www.targetscan.org). Seed sequences are shown in the red box; homologous seed nucleotides are shown in red letters and mismatches are shown in black letters in the seed sequence box. “-” indicates a gap in the sequence. In the circRNA–miRNA alignments in (**B**), G–U wobbles are underlined. NA, not available in databases.

**Figure 3 ijms-24-04549-f003:**
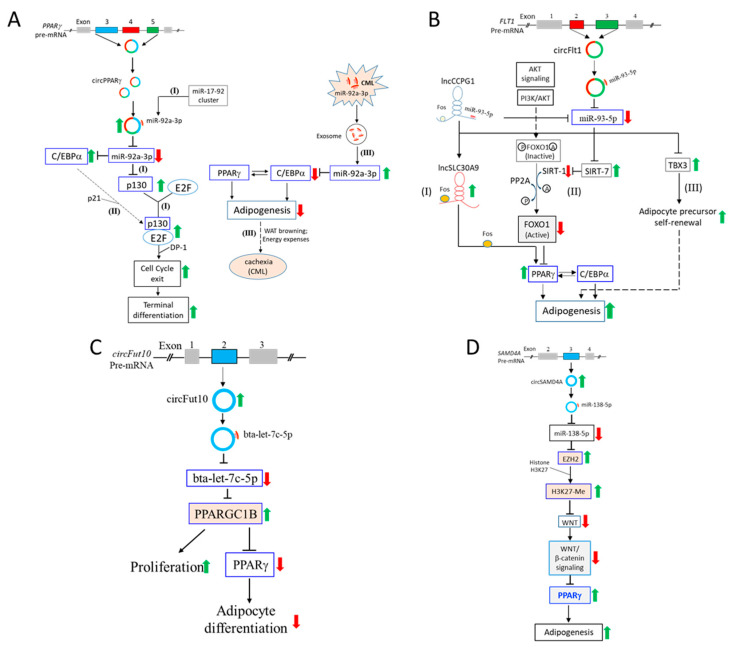
Selected circRNA- and miRNA-mediated post-transcriptional regulation of signaling and biochemical pathways in preadipocyte differentiation. (**A**) CircPPARγ sponges miR-92a-3p to regulate C/EBPα and p130/Rb2. Depicted are the impacts of circPPARγ sponging of miR-92a-3p on adipogenesis via p130 (left panel, route I) and C/EBPα (left panel, routes I and II). Cancer-associated cachexia in chronic myeloid leukemia (CML) through adipogenesis is also shown (right panel, route III). (**B**) CircFLT1 sponges miR-93-5p to regulate SIRT-7 and TBX3. Route (I) involves lncCCPG1 and lncSLC30A9 and the c-Fos protein; route (II) depicts SIRT-1 and -7 and the FOXO1 pathway targeting PPARγ; route (III) illustrates TBX3 regulation of precursor renewal prior to differentiation in the early stage of adipogenesis. (**C**) The circFUT10-let-7c-5p-PCG1β regulatory pathway. Negative modulation of preadipocyte differentiation by circFUT10 is shown. (**D**) The circSAMD4A-miR-138-5p-EZH2 pathway. In this pathway, circSAMD4A sponges miR-138-5p to modulate EZH2-induced histone methylation. See text for further description and for sources based on which the schemes are constructed. In all subfigures, thin and blunted arrows indicate positive and negative regulation, respectively, in normal cells; thick upward-pointing green and red downward-pointing arrows indicate the modulatory effects of the circRNAs on the proteins in the pathway accumulating to the modulation of preadipocyte differentiation in adipogenesis; arrowheads with dashed lines indicate other multi-step pathways, the details of which are not indicated.

**Figure 4 ijms-24-04549-f004:**
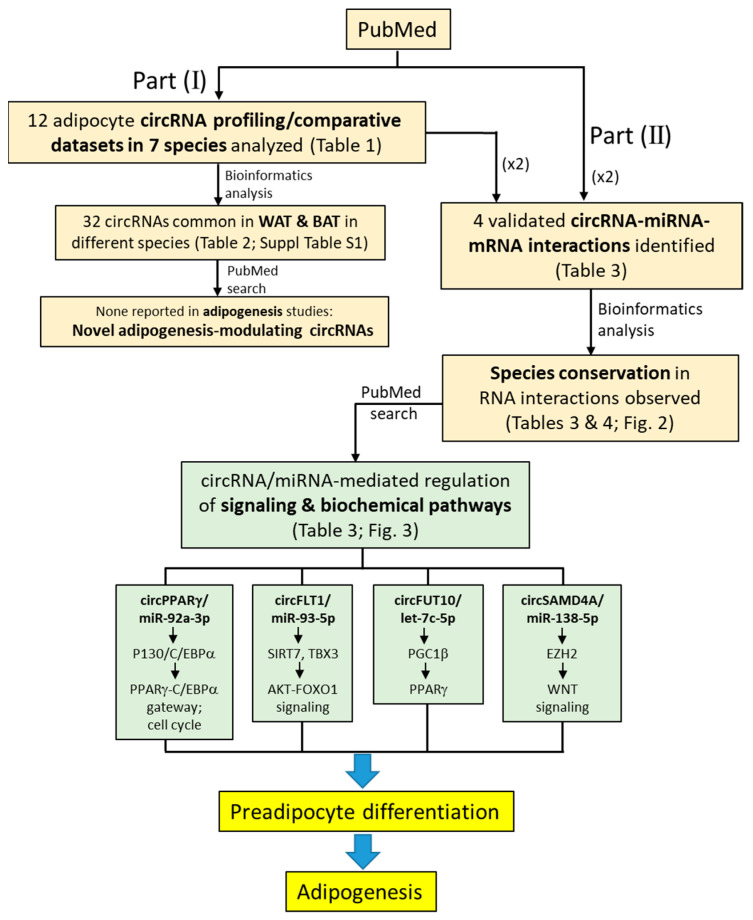
Pathway of analysis in the identification of circRNAs, miRNAs, proteins and modulated pathways leading preadipocyte differentiation in adipogenesis. Part (I): Identification of circRNAs using 12 datasets derived in circRNA profiling and comparative studies in different species. Twenty-three circRNAs common to two or more WATs and BATs in different species are identified, none of which has not been reported in adipogenesis studies. Part (II): Identification of selected circRNAs, which are validated in circRNA–miRNA–mRNA interactions, and which show species conservation in the RNA interacting sequences. The circRNA/miRNA-mediated signaling and biochemical pathways leading to adipogenesis differentiation are also analyzed. See text for details.

**Table 1 ijms-24-04549-t001:** Adipocyte circRNA profiling papers and datasets in relation to preadipocyte differentiation and comparative studies.

No.	Species	Adipose Tissue Profiled	Platform ^a^ (RNA Prep’n)	Profiling Dataset Availability	Number in circRNA		Reference
					Total	DE (Up/Down) on Different’n/Comparison	
(A) Preadipocyte Differentiation Studies							
A1	Human	VAT	circRNA microarray (total RNA)	Yes	NA	4080 (2215/1865)	[50]
A2	Mouse	WAT (stromal cells)	RNA-seq (circRNA-enriched)	Yes	3771	41 (1.09%) (28/13)	[51]
A3	Mouse	BAT (stromal cells)	RNA-seq (circRNA-enriched)	Yes	3869	117 (3.02%) (77/40)	[52]
A4	Yak	SAT	RNA-seq (circRNA-enriched)	Yes	7203	136 (1.89%) (92/44)	[53]
A5	Pig	SAT	RNA-seq	Yes (chrom’l sites only)	2172	297 (13.67%) (Also, lncRNA and mRNA)	[54]
**(B) Differentiation and Comparative Studies**							
B1	Human	VAT	circRNA microarray (circRNA-enriched)	Yes	NA	Obese vs. lean: 244 (143/101)	[47]
B2	Human	VAT and SAT	RNA-seq	Yes	6925	NA	[49]
B3	Mouse	Epididymal and inguinal fat (WAT)	RNA-seq	Yes	2380	NA	[49]
B4	Pig	SAT	RNA-seq (circRNA-enriched)	No	29,763 (combined)	Two breeds: 275 (70/205)	[55]
B5	Rat	SAT (stromal cells)	RNA-seq	No	Young: 4860 Old: 4952	Young vs. old: 67 (1.38%) (33/34)	[56]
B6	Cattle	SAT	RNA-seq	No	Calf: 4337 Adult: 5465	Calf vs. adult: 307 (7.08%) (156/151)	[48]

^a^ RNA-seq, whole transcriptome RNA sequencing. The RNA preparations used were either rRNA-free, or “circRNA-enriched” RNA preparations in which both rRNA and linear RNA species were depleted. VAT and SAT, visceral and subcutaneous adipose tissues; WAT and BAT, white and brown adipose tissues; DE, differential expression; NA, not available.

**Table 2 ijms-24-04549-t002:** Common circRNAs in WAT and BAT profiling datasets of different species.

Dataset No.	Dataset ^a^	No. of DE circRNAs		No. of Overlapping circRNAs in WAT-1 (Percentage) ^a^	Reference
		Upregulated	Downregulated		
1	Mouse WAT-1	28	13	41 (100%)	[51]
2	Mouse WAT-2	20	5	25 (61.0%)	[49]
3	Human WAT	18	5	23 (56.1%)	[49]
4	Yak WAT	14	5	19 (46.3%)	[53]
5	Mouse BAT	25	7	32 (78.0%)	[52]

^a^ The mouse WAT dataset 1 of reference [51] is used as the reference for comparison with other datasets. See Appendix A for more details on the circRNAs identified.

**Table 3 ijms-24-04549-t003:** Validated circRNA–miRNA–mRNA connections in preadipocyte differentiation.

CircRNA			Presence in WAT/BAT Dataset ^a^	miRNA (Family)	mRNA (Gene Symbol)	Reference
Host Gene	CircRNA (circBase ID) (Species)	Exons				
PPARγ(Peroxisome Proliferator-Activated Receptor-gamma)	bta_circ_Pparγ (bta_circ_0010660) (Bovine)	3-5	h, m, y	miR-92a-3p/(mir-25)	p130/Rb2RB (Transcriptional Corepressor Like 2) C/EBPα (CCAAT Enhancer Binding Protein Alpha)	[57,58,59,60]
FLT1(Fms-Related Receptor Tyrosine Kinase 1	bta_circ_Flt1 (bta_circ_002673) (Bovine)	2-3	y	miR-93-5p (mir-17) (and lncCCPG1 and lncSL30A9)	SIRT7 (Sirtuin 7) TBX3 (T-Box Transcription Factor 3)	[61,62]
FUT10(Fucosyltransferase 10)	bta_circ_Fut10 (NA) (Bovine)	2	h, y	let-7c-5p (let-7)	PGC1β/PPARGC1B (PPARγ Coactivator 1-β)	[48]
SAMD4A(Sterile Alpha Motif Domain Containing 4A)	hsa_circ_SAMD4A (hsa_circ_0004846) (Human)	3	h, m, y	miR-138-5p (mir-138)	E2H2 (Enhancer of Zeste 2 Polycomb Repressive Complex 2)	[47]

^a^ h, human, m, mouse, y, yak. The datasets are as in Table 2.

**Table 4 ijms-24-04549-t004:** Species conservation of miRNA seed sequences and in circRNA–miRNA and miRNA–mRNA interactions.

circRNA	miRNA	mRNA	miRNA Seed Sequence Conservation		
			miRNA in Different Species ^a^	cirRNA–miRNA Interaction ^b^	miRNA–mRNA Interaction ^a^
circPPARγ	miR-92a-3p	C/EBPα p130/RB2	All 7/7	All 7/7	C/EBPα: All 7/7 except chicken: 6/7 p130/RB2: All 7/7 except rat: 6/7
circFLT1	miR-93-5p	SIRT7 TBX3	All 7/7 (NA for pig and chicken)	All 7/7 except mouse: 6/7	SIRT7: All 8/8 except rat: 7/8 (NA for chicken) TBX3: All 8/8
circFUT10	let-7c-5p	PGC1β	All 6/6	All 6/6 except mouse: 4/6	Seed I: All 7/7 Seed II: All 7/7, except pig: 6/7
circSAMD4A	miR-138-5p	EZH2	All 7/7	All 6/6	All 7/7, except chicken: 6/7

Details of seed sequence alignments are as shown in Figure 2. Seed sequences are compared in ^a^ human, mouse, rat, pig, cow and chicken, or in ^b^ human, mouse and bovine. NA, not available.

## Data Availability

Not applicable.

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
