# Peer review of "Circular RNA- and microRNA-Mediated Post-Transcriptional Regulation of Preadipocyte Differentiation in Adipogenesis: From Expression Profiling to Signaling Pathway"

_ijms, 2023, doi:10.3390/ijms24054549_

Round 1
Reviewer 1 Report
This is interesting manuscript. Content is relevant to the Title. Conclusions are clear.
This manuscript could be appropriate for publishing.
Author Response
No replies needed. Thank you.

Reviewer 2 Report
Dear Authors, I see that you presented quite a nice review concerning post-transcriptional regulation of adipogenesis by circRNA-miRNA-miRNA connections. Here is presented valuable consensus information gathered in one place, which could be helpful for further study considering adipogenesis regulation. In my opinion, the manuscript is well written, I just found small punctuation mistakes to check throughout the manuscript.
I propose minor revisions; below are suggestions:
line 110-112, there should be added that miRNA can also be generated from host genes
Paragraph - 123-129 here can be added what should be specific (which feature) in the transcript to be circRNA generated from this transcript because is known that circRNA are sometimes generated from mRNA
Table 1. here, in some cases, is indicated that RNA-seq or microarray was circRNA- enriched or total RNA. Please add the information where is written only "RNA-seq", that there were depleted RNA, total RNA or poly-A selected libraries used, to be consistent.
Author Response
Dear Authors, I see that you presented quite a nice review concerning post-transcriptional regulation of adipogenesis by circRNA-miRNA-miRNA connections. Here is presented valuable consensus information gathered in one place, which could be helpful for further study considering adipogenesis regulation. In my opinion, the manuscript is well written, I just found small punctuation mistakes to check throughout the manuscript.
I propose minor revisions; below are suggestions:
line 110-112, there should be added that miRNA can also be generated from host genes
Authors’ response:
Accepting reviewer’s kind suggestion, we have added the a new statement and two references in the revision (lines 114-116) as follows:
“It is noteworthy that many miRNA genes are located within the intron sequences of many protein-coding genes, a genomic framework shaped by evolution [40, 41].”
[40] França GS, Vibranovski MD, Galante PA. (2016) Host gene constraints and genomic context impact the expression and evolution of human microRNAs. Nat Commun. 7, 11438.
[41] Boivin V, Deschamps-Francoeur G, Scott MS. (2018) Protein coding genes as hosts for noncoding RNA expression. Semin Cell Dev Biol. 75:3-12.
Paragraph - 123-129 here can be added what should be specific (which feature) in the transcript to be circRNA generated from this transcript because is known that circRNA are sometimes generated from mRNA
Authors’ response:
We have added a statement to illustrate some structural features that contribute to circRNA biogenesis in the backsplicing process as follows (lines 129-133):
“Backsplicing requires the canonical pre-mRNA spliceosomal machinery, recognizing the same consensus intron-exon GU-AG junctions. The process is also modulated by pairing of intronic repetitive sequences, such as Alu, and contribution of cis complementary sequences and trans-acting protein factors [45, 46].”
[45] Wilusz JE (2018) A 360 degree view of circular RNAs: from biogenesis to functions. Wiley Interdiscip Rev RNA. 2018 Jul; 9(4): e1478.
[46] Li X, Yang L, Chen LL (2018) The biogenesis, functions, and challenges of circular RNAs. Mol Cell 71, 428-442.
Table 1. here, in some cases, is indicated that RNA-seq or microarray was circRNA- enriched or total RNA. Please add the information where is written only "RNA-seq", that there were depleted RNA, total RNA or poly-A selected libraries used, to be consistent.
Authors’ response:
There were only two kinds of RNA preparation used in the RNA-seq profiling: preparations depleted of rRNA, i.e. rRNA-free, or preparation depleted of both rRNA and linear RNA. We have revised the table footnote to clarify this issue as follows (lines 185-186):
“aRNA-seq, whole transcriptome RNA sequencing. The RNA preparations used in RNA-seq profiling were either rRNA-free, or “circRNA-enriched” RNA preparations in which both rRNA and linear RNA species were depleted.”

Reviewer 3 Report
In their review Huang and Choo discussed how circRNAs and microRNAs modulate post-transcriptional expression of targeted mRNAs impacting adipogenesis signaling pathways. In particular they analyzed four circRNA-miRNA-mRNA interactions together with the downstream pathways involved in preadipocyte differentiation in different species.
The review is quite original and interesting, and data have been thoroughly searched in literature. Only some minor concerns should be addressed by the Authors, before a publication on International Journal of Molecular Sciences can be granted:
MINOR REVISIONS:
· In the “Introduction” a picture of preadipocytes differentiation and some more informations about it may be useful;
· Line 44: dysfunctional WAT is also associated with many types of cancer;
· Paragraph 1.2.: you need to explain more specifically what are circRNAs and miRNAs, how they are generated and how they can act and the difference between them and lncRNAs;
· “It is noted here in passing” in line 102 does not make sense;
· Table 1 and fig.2: please make the letters (A), (B) and (C) more visible;
· Line 222: correct the phrase “producing leaner persons”;
· Fig.2: enlarge the single figures and add the words “alignment” in fig.B and “conservation” in fig.C;
· Line 315: the mismatches in circ-FUT10-let-7c-5p are not underlined;
· Fig.3: please make the graphs clearer (activation and inhibition are not well understandable);
· Line 376: route (II) is missing;
· Line 420-426: please explain better this concept because it does not make sense;
· Always write latin words in italics;
· English editing and typinng errors’ check is mandatory.
Overall. MINOR REVISIONS are required.
Author Response
Comments and Suggestions for Authors
In their review Huang and Choo discussed how circRNAs and microRNAs modulate post-transcriptional expression of targeted mRNAs impacting adipogenesis signaling pathways. In particular they analyzed four circRNA-miRNA-mRNA interactions together with the downstream pathways involved in preadipocyte differentiation in different species.
The review is quite original and interesting, and data have been thoroughly searched in literature. Only some minor concerns should be addressed by the Authors, before a publication on International Journal of Molecular Sciences can be granted:
MINOR REVISIONS:
In the “Introduction” a picture of preadipocytes differentiation and some more informations about it may be useful.
Authors’ response:
In the revision, we have added new statements and a new reference as follows:
“Adipose tissue mass may expand via increasing the size of the constituent adipocyte cells. On the other hand, adipogenesis is an adipocyte biogenesis process in which new adipocytes are generated from multipotent progenitor stem cells. Adipogenesis begins with the progenitor cells being committed to become preadipocytes, which undergo growth arrest, followed by preadipocyte differentiation into mature adipocytes [1 , 2].
[2] Gregoire FM et al. (1998) Understanding adipocyte differentiation. Physiol Rev. 78, 783-809.
Line 44: dysfunctional WAT is also associated with many types of cancer
Authors’ response:
Agreed. We thought the phrase “ ... among other human conditions ...” sufficed. In the revision, we have added “... and cancers ...” (line 45), and citing two reviews by Daas et al. (2018) and Nunn et al. (2022).
[8] Daas SI, Rizeq BR, Nasrallah GK. (2018) Adipose tissue dysfunction in cancer cachexia. J Cell Physiol. 234, 13-22.
[9] Nunn ER, Shinde AB, Zaganjor E. (2022) Weighing in on adipogenesis. Front Physiol. 13:821278.
Paragraph 1.2.: you need to explain more specifically what are circRNAs and miRNAs, how they are generated and how they can act and the difference between them and lncRNAs.
Authors’ response:
The generation and actions of circRNA and miRNA are clearly depicted in Figure 1, and briefly explained in the first paragraph of Section 1.2 (lines 79-92). A brief mention of lncRNA is found in the second paragraph of Section 1.2 (lines 103-107). Hence, we do not think revision is necessary.
“It is noted here in passing” in line 102 does not make sense.
Authors’ response:
In English expression, we mean “lncRNA is briefly mentioned here but without giving details”. Since lncRNA is not the subject of this review, we don’t see why it does not make sense, and, hence, no revision is made.
Table 1 and fig.2: please make the letters (A), (B) and (C) more visible.
Authors’ response:
The sizes of (A) (B) (C) lettering in Table 1 and Fig. 2 should be the same as the lettering of the subtitles. If we make the (A) (B) (C) lettering larger to be “more visible”, that will make lettering in the table/figure disharmonized.
Line 222: correct the phrase “producing leaner persons”;
Authors’ response:
Revised to “slimmer” (line 227).
Fig.2: enlarge the single figures and add the words “alignment” in fig.B and “conservation” in fig.C;
Authors’ response:
The subfigures are slightly enlarged but still keeping the two/one format. The production team of the publisher may want to change the format to one dataset per row.
Adding “alignment” and “conservation” in all subtitles would make the titles long and repetitious. (A) and (B) is meant to demonstrate conservation between species while (C) is meant to illustrate alignment of the seed sequences between the miRNAs and the 3’-UTR of the targeted transcripts.
However, we have added “alignment” in the Figure 2 title as follows: “Figure 2. Cross-species alignment and conservation of the ...” (line 309).
Line 315: the mismatches in circ-FUT10-let-7c-5p are not underlined;
Authors’ response:
Yes, our mistake. “Underlined” is now amended to “in black letters” (line 325) for mismatch; underlined and red letters are meant for wobble bases.
Fig.3: please make the graphs clearer (activation and inhibition are not well understandable).
Authors’ response:
Gene activation and inhibition are well-defined molecular terms. Sorry we don’t see how we could explain the terms clearer. In the Figure 3 legends, we use “... positive and negative regulation ...” (line 391).
Line 376: route (II) is missing.
Authors’ response:
Thanks for picking up the error. We have now inserted “(II)” (line 385).
Line 420-426: please explain better this concept because it does not make sense;
Authors’ response:
We have slightly revised the original paragraph by separating acetylation and AKT phosphorylation, and have added a new statement explaining how circFlt-1 sponging of miR-93-5p leads to activation of adipogenesis in route (II):
Lines 430-439
“In knockout mice, SIRT-7 is shown to deacetylate and, hence, activate another SIRT protein, SIRT-1, in the preadipocyte differentiation process (Figure 3B, route II) [77]. On the other hand, FOXO1 (Forkhead Box O1), previously inactivated upstream by being acetylated and also phosphorylated by AKT signaling, is now being re-activated by deacetylation by SIRT-1 and dephosphorylation by PP2A (protein phosphatase 2). Subsequently, the activated FOXO1 protein binds to the PPARγ promoter to block PPARγ expression in cis, or interacts with PPARγ in trans, to deplete PPARγ for utilization in adipogenesis (Figure 3B, route II) [16, 78-81]. When miR-93-5p is sponged by circFlt-1, expression of SIRT-7 is up-regulated, suppressing SIRT-1 expression to prevent deacetylation and re-activation of FOXO1, thus, allowing PPARg to participate in adipogenesis.”
Always write latin words in italics.
Authors’ response:
Agreed. And we have checked.
English editing and typinng errors’ check is mandatory.
Authors’ response:
We have checked and made the necessary amendments.
Overall. MINOR REVISIONS are required.
Authors’ response:
Thank you.

Reviewer 4 Report
The review article by Huang and Choo comprehensively summarize the role of circular RNA and microRNA in adipogenesis. They utilized dataset from published papers indexed in Pubmed, covering several species including mice and human.
I would like to address minor comments as below:
1. Authors stated that 27 circRNA are commonly expressed in WAT and BAT differentiation (line 246-247). However, in the Figure 4, it is written 23 common circRNA in WAT and BAT. I am also curious about other circRNA which are not common between WAT and BAT, they may have specific role for WAT or BAT differentiation.
2. As BAT or brown/beige adipocytes are positively correlated with metabolic function, as I wrote in the previous point, discussing circRNA or miRNA, which increase the browning or drive the differentiation towards brown/beige program, will be interesting.
3. in line 444-445, authors wrote "in adipogenesis, however, PGC1B acts as a PPARgamma repressor in competing with other PPARgamma activators to block PPARgamma action in adipocyte differentiation." Please provide the reference for this sentence.
4. in line 446-447, authors wrote "It is also noteworthy that PGC1β has been shown to be activated by PRDM16 (PR/SET Domain 16 Protein) in fate determination of brown fat adipogenesis [82]." This is somehow contradict with the previous sentence that PGC1β suppressed PPARgamma.
